# Trends of in-hospital cardiac arrests in a single tertiary hospital with a mature rapid response system

Hohyung Jung[1], Ryoung-Eun Ko[1], Myeong Gyun Ko[2], Kyeongman Jeon[1,3]*

**1** Department of Critical Care Medicine, Samsung Medical Center, Sungkyunkwan University School of Medicine, Seoul, Korea, **2** Intensive Care Unit Nursing Department, Samsung Medical Center, Sungkyunkwan University School of Medicine, Seoul, Korea, **3** Division of Pulmonary and Critical Care Medicine, Department of Medicine, Samsung Medical Center, Sungkyunkwan University School of Medicine, Seoul, Korea

☯ These authors contributed equally to this work.

* kjeon@skku.edu

**Data Availability Statement:** The data that support the findings of this study cannot be shared publicly because of legal and ethical restrictions, however, data can be made available to interested researchers upon request. Requests can be

## Abstract

### Background

Most studies on rapid response system (RRS) have simply focused on its role and effectiveness in reducing in-hospital cardiac arrests (IHCAs) or hospital mortality, regardless of the predictability of IHCA. This study aimed to identify the characteristics of IHCAs including predictability of the IHCAs as our RRS matures for 10 years, to determine the best measure for RRS evaluation.

### Methods

Data on all consecutive adult patients who experienced IHCA and received cardiopulmonary resuscitation in general wards between January 2010 and December 2019 were reviewed. IHCAs were classified into three groups: preventable IHCA (P-IHCA), non-preventable IHCA (NP-IHCA), and inevitable IHCA (I-IHCA). The annual changes of three groups of IHCAs were analyzed with Poisson regression models.

### Results

Of a total of 800 IHCA patients, 149 (18.6%) had P-IHCA, 465 (58.1%) had NP-IHCA, and 186 (23.2%) had I-IHCA. The number of the RRS activations increased significantly from 1,164 in 2010 to 1,560 in 2019 (P = 0.009), and in-hospital mortality rate was significantly decreased from 9.20/1,000 patients in 2010 to 7.23/1000 patients in 2019 (P = 0.009). The trend for the overall IHCA rate was stable, from 0.77/1,000 patients in 2010 to 1.06/1,000 patients in 2019 (P = 0.929). However, while the incidence of NP-IHCA (P = 0.927) and I-IHCA (P = 0.421) was relatively unchanged over time, the incidence of P-IHCA decreased from 0.19/1,000 patients in 2010 to 0.12/1,000 patients in 2019 (P = 0.025).

directed to the secretariat of the department of critical care medicine (contact via hyerim37. jeong@samsung.com) or the corresponding author. Sharing of individual data is available only when the corresponding author requests to the Research Data Review Committee and obtains approval from the committee.

**Funding:** This work was supported by a Samsung Medical Center grant (SMO1200901). The funders had no role in study design, data collection and analysis, decision to publish, or preparation of the manuscript.

**Competing interests:** The authors have declared that no competing interests exist.

## Conclusions

The incidence of P-IHCA could be a quality metric to measure the clinical outcomes of RRS implementation and maturation than overall IHCAs.

## Introduction

Despite the advances in the management of in-hospital cardiac arrest (IHCA) over the past decade, IHCA remains associated with poor outcomes [1,2]. However, while it is frequently preceded by a more gradual, possibly treatable, decline [3–6], many cases of IHCAs are considered preventable based on retrospective reviews [3,5–7]. To reduce the incidence of IHCA, a rapid response system (RRS) was designed to identify early signs of clinical deterioration and activate a specialized team of caregivers [8]. Most recently, the International Society of RRS recommends that hospitals collect data on IHCAs and their potential predictability [9].

A cardiac arrest is treated with cardiopulmonary resuscitation (CPR), which is lifesaving for patients who have a history of acute, potentially reversible illness. However, this is not effective if cardiac arrest occurs in patients in fatal conditions with terminal illness. "Do not attempt CPR" (DNACPR) decisions allow resuscitation to be withheld when the chance of success is little or when the burdens of CPR outweigh the benefits [10]. Nevertheless, physicians are occasionally faced with patients requesting full resuscitation against medical advice. More commonly, neither patients nor their family members make such a request, but physicians simply presume that providing CPR comports with the patient's wishes [11,12]. Therefore, performing CPR on all IHCAs, regardless of the severity of the underlying illness and end-of-life medical decision, may be inappropriate [13]. In contrast, the implementation of RRS also increases the likelihood of DNACPR [14], which could be partially attributed to the reduced IHCA cases after RRS implementation but with a lesser impact on hospital mortality.

However, most studies on RRS have simply focused on its role and effectiveness in reducing IHCA or hospital mortality, regardless of the type of IHCA [8]. As a result, it has been difficult to determine the overall rate of potentially avoidable IHCA and if this rate is changing with implementation and maturity of RRS. Therefore, this study aimed to identify the rates and characteristics of IHCAs including predictability of the IHCAs as our RRS matures for 10 years, to determine the best measure for RRS evaluation.

## Methods

### Study design, setting, and participants

This retrospective observational study included all consecutive patients who experienced IHCA and received CPR in general wards at Samsung Medical Center, Seoul, South Korea between January 2010 and December 2019; this university affiliated tertiary referral hospital has a 1,989-bed capacity with a hospital-wide medical emergency team (MET) for the RRS. To address the primary research question of whether characteristics of IHCAs in hospitalized adult patients is associated with maturity of our MET over 10 years, we reviewed the clinical data of all treated IHCAs through the electronic medical records. This study was approved by the institutional review board of the Samsung Medical Center and performed in compliance with Helsinki declaration. The institutional review board waived the requirement for informed consent due to the observational nature of the research. Additionally, the patients' information was anonymized and de-identified prior to analysis.

## Operation of the RRS

The hospital-wide MET at the Samsung Medical Center was introduced at the beginning of March 2009, consisted of either fellows that were training in critical care or senior residents in internal medicine [15–18]. Since March 2013, the MET consists of dedicated intensivist physicians, including critical care fellows and attending intensivists, which provide round-the-clock coverage. All hospital medical personnel were presented with information about the MET and educated to prevent inadequate clinical assessments of patient deterioration that cause delays in the MET activation. Before implementation of the automated system, physicians and nurses directly contacted the MET using a dedicated phone number when a patient met any single criterion (Table 1). Activation was also allowed when the medical staff was concerned about changes in their patient's clinical condition, even in the absence of physiological disorders that meet the criteria. In August 2016, the MET initiated an automated alert and activation system for all ward patients using a modified early warning score (MEWS) [19]. The MEWS was automatically calculated using five physiological parameters (systolic blood pressure, heart rate, respiratory rate, body temperature, and level of consciousness) when nurses records the patient's vital signs on the electronic medical record. Patient vital signs were recorded at the bedside immediately after measurement using a laptop or portable device whenever possible. MEWS was automatically updated with each new vital sign recorded. The frequency of measuring vital signs was made according to the order of the attending physician, but vital signs were usually measured at least four times a day and more often when the patient's clinical condition changed. If the MEWS was 7 or higher, an automated alert was sent to MET as a text message in real-time, 24-hours a day, 7 days a week, and MET was automatically activated. Calls for MET activation were available for all patients regardless of do-not-resuscitate status during the study period.

When activated, the MET is expected to arrive within 10 min, complete patient assessments within 30 min, and order diagnostic tests and therapeutic treatments relevant to the patient's condition. In certain clinical problems requiring specialized expertise, other teams such as acute myocardial infarction team, acute stroke team, and acute care surgery team also can be activated. Following assessment and initial treatment, an individual treatment plan is created for each patient, and a joint decision is made about whether to transfer the patient to the intensive care unit (ICU). The issue of limitation on medical intervention and end-of-life care can also be discussed at this point. After assessment and therapeutic interventions by the MET,

**Table 1. Calling criteria for the medical emergency team.**

| Airway and breathing | • Acute respiratory distress: respiratory rate $\geq$ 30 breaths/min<br>• Acute hypoxia: oxygen saturation derived from pulse oximetry < 90% for 5 min, despite previous oxygen administration<br>• Acute hypercapnia and acute acidosis: arterial carbon dioxide pressure > 50 mmHg and pH < 7.3<br>• Upper airway obstruction: stridor or use of respiratory accessory muscle |
|---|---|
| Circulation | • Unexplained hypotension: systolic blood pressure < 90 mmHg<br>• Acute chest pain<br>• Bradycardia or tachycardia: heart rate < 50 beats/min or > 130 beats/min<br>• Arrhythmia with symptom |
| Neurology | • Sudden mental change or unexplained agitation<br>• Seizure |
| Other | • Bedside concern about overall deterioration |

patients who are considered to require treatment and monitoring that cannot be provided outside of the ICU are transferred to the ICU, while patients in a stable condition remain on the general ward. For patients who are not fully stabilized and require intensive monitoring but are able to manage with lower levels of care than in the ICU, admission to the unit is decided on a case-by-case basis. The MET determined the patient's disposition and shared information about the advance care plan with the primary care team. Discussion about end-of-life care and ceiling of care were also included in certain patient populations. The decisions about completion of intervention and disposition were left to the judgment of each member of the MET without specific criteria, but the decision making generally followed international guidelines [20].

Details of all MET calls were recorded as soon as possible after the event by a member of the team and were entered into a registry. This recorded patient demographics, reasons for the MET activation based on calling criteria, time of the first documented physiological disorder, modified early warning score, time of the MET activation and deactivation, vital signs at the time of the MET activation and deactivation, interventions delivered by the MET, and the final outcomes including the patient's disposition after the clinical episode [21,22]. These data were supplemented on the next day with a retrospective review of hospital medical records before registration for quality control of registry data.

### Definitions

All IHCAs were classified into three groups [23,24]: Preventable IHCA (P-IHCA) was defined as a cardiac arrest with preexisting signs of acute physiologic disturbance that fulfilled the MET activation criteria from 8 hours to 30 minutes before arrest. Non-preventable IHCA (NP-IHCA) was defined as a cardiac arrest that occurred within 8 hours after admission, or without any record of vital signs within 8 hours before arrest, or within 30 minutes after drug administration or procedures, or from unexpected lethal arrhythmia; this includes cardiac arrest that occurred within 30 minutes after MET activation. Inevitable IHCA (I-IHCA) was defined as IHCA in patients who had already requested a DNACPR order or were in terminal health conditions. Cases that were difficult to be classified were resolved by consensus of the MET team.

### Statistical analysis

The MET dose was calculated by the number of MET calls per annum divided by the total number of discharged patients per year, represented as cases per 1,000 patients. In addition, each rates of IHCAs according to the classification or in-hospital mortality was calculated by the number of IHCA or in-hospital mortality per annum divided and represented as cases per 1,000 patients. Data were presented as number and percentages. The annual changes of the numbers and rates were analyzed with Spearman correlation analysis and Poisson regression models, respectively. All data were analyzed using SPSS version 22 (IBM Corp., Armonk, NY).

### Results

During the 10-years study period, 843,180 patients were admitted to Samsung Medical Center and a total of 824 consecutive IHCAs were recorded for adult patients. After excluding duplicated CPRs for the same IHCAs (n = 24), a total of 800 treated IHCAs with CPR on the general ward were retrieved for the primary analysis (0.95/1,000 patients). The baseline characteristics of 800 IHCA patients are given in Table 1. There were 467 (58.4%) male, and the median age was 64.5 (IQR, 53.0–74.0) years. Malignant disease (47.8%) and cardiovascular disease (26.8%) were the most frequent comorbidities. The median hospital admission day before arrest was

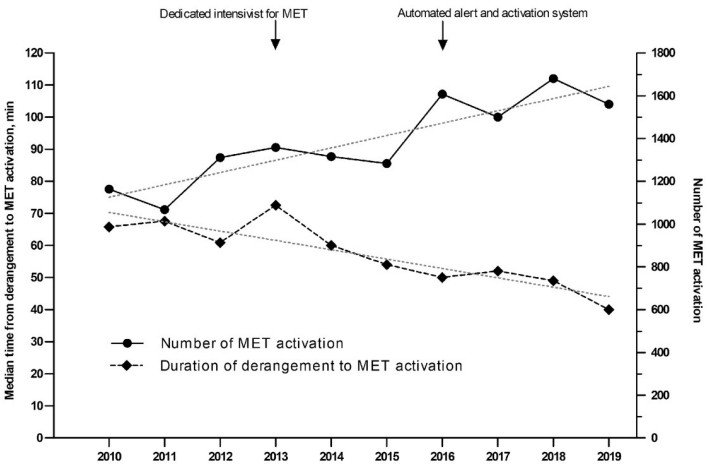

**Fig 1. Medical emergency team (MET) maturation per year since 2010.** The circles and lines represent of the number of MET activations. The number of MET activations increased from 1,164 in 2010 to 1,560 in 2019 (P for trend = 0.002). The diamonds and dotted lines represent the time from derangement to MET activation. The time from derangement to MET activation decreased from 66 minutes in 2010 to 40 minutes in 2019 (P for trend < 0.001). MET, medical emergency team.

9.7 (IQR 4.0–22.7) days. The most IHCA was occurred during weekdays (68.9%) and half of events occurred at daytime. Common location was general ward (58.3%), followed by monitoring room (36.4%). Cardiovascular (29.5%) was common cause of arrest, and 50.6% initially presented with pulseless electrical activity rhythm. Extracorporeal cardiopulmonary resuscitation was applied to 22 (7.2%) patients.

As MET matured, the number of the MET activations increased significantly from 1,164 in 2010 to 1,560 in 2019 (P = 0.002) (Fig 1). In addition, the median time from derangement to MET activation decreased from 66 minutes in 2010 to 40 minutes in 2019 (P < 0.001).

Of the 800 IHCA patients, 149 (18.6%) had P-IHCA, 465 (58.1%) had NP-IHCA, and 186 (23.2%) had I-IHCA (Table 2).

Among 252 patients (31.5%) survived at discharge, only 121 (15.1%) patients were managed by our MET within 24 hours before the IHCA. However, the MET dose was relatively unchanged over time (P = 0.531), since the number of admitted patients increased from 72,468 in 2010 to 100,788 in 2019 (P < 0.001). Finally, in-hospital mortality rate was significantly decreased from 9.20/1,000 patients in 2010 to 7.23/1,000 patients in 2019 (P = 0.004) (Table 3).

The trend for the overall IHCA rate was stable from 0.77/1,000 patients in 2010 to 1.06/1,000 patients in 2019 (P for trend = 0.720) (Fig 2). However, the incidence of NP-IHCA (P for trend = 0.382) and I-IHCA (P for trend = 0.054) was relatively unchanged over time, while that of P-IHCA decreased from 0.19/1,000 patients in 2010 to 0.12/1,000 patients in 2019 (P for trend = 0.006) (Fig 2).

## Discussion

This study investigated the change of overall and various type of IHCAs following RRS implementation and maturation over 10 years. The major finding is that the RRS call had been increased significantly as the RRS had been maturated, and the P-IHCA and in-hospital mortality were decreased significantly. However, the overall rate of IHCAs did not change significantly during the study period.

**Table 2. Clinical characteristics of in-hospital cardiac arrests (N = 800).**

| Variables | No. of patients or median (IQR) |
|---|---|
| Sex, male | 467 (58.4) |
| Age, year | 64.5 (53.0–74.0) |
| Medical department admission | 583 (72.9) |
| Comorbidities | |
| Cardiovascular disease | 214 (26.8) |
| Respiratory disease | 36 (4.5) |
| Malignant disease | 382 (47.8) |
| Central nervous system | 63 (7.9) |
| Hepatobiliary disease | 37 (4.6) |
| Chronic kidney disease | 42 (5.3) |
| Documented treatment limitation | 103 (12.9) |
| Hospitalization prior to arrest, day | 9.7 (4.0–22.7) |
| IHCA day and time period | |
| Weekday | 551 (68.9) |
| Daytime hours (8:00 ~ 18:00) | 368 (46.0) |
| Location of arrest | |
| General ward | 466 (58.3) |
| Monitoring room | 291 (36.4) |
| Procedure room | 18 (2.3) |
| Others | 25 (3.1) |
| Monitored patients | 543 (67.9) |
| MET activation within 24 hours of IHCA | 121 (15.1) |
| Witnessed arrest | 683 (85.4) |
| Presumed reason for arrest | |
| Cardiovascular arrest | 236 (29.5) |
| Respiratory arrest | 352 (44) |
| Hypovolemic shock | 60 (7.5) |
| Sepsis | 43 (5.4) |
| Brain injury | 10 (1.3) |
| Anaphylaxis | 8 (1.0) |
| Unknown | 91 (11.4) |
| Initial rhythm | |
| Shockable | 147 (18.4) |
| Pulseless electrical activity | 405 (50.6) |
| Asystole | 216 (27.0) |
| Not available | 32 (4.0) |
| Extracorporeal cardiopulmonary resuscitation | 58 (7.2) |
| Classification of IHCA | |
| P-IHCA | 149 (18.6) |
| NP-IHCA | 465 (58.1) |
| I-IHCA | 186 (23.2) |
| Survivor at hospital discharge | 252 (31.5) |

No., number; IQR, interquartile range; IHCA, in-hospital cardiac arrest; P-IHCA, preventable in-hospital cardiac arrest; NP-IHCA, non-preventable in-hospital cardiac arrest; I-IHCA, inevitable in-hospital cardiac arrest; MET, medical emergency team.

**Table 3. Annual trend of the MET activation and in-hospital mortality.**

|  | 2010 | 2011 | 2012 | 2013 | 2014 | 2015 | 2016 | 2017 | 2018 | 2019 | P for trend |
|---|---|---|---|---|---|---|---|---|---|---|---|
| Number of MET activation | 1,164 | 1,068 | 1,311 | 1,359 | 1,316 | 1,284 | 1,608 | 1,500 | 1,680 | 1,560 | 0.002 |
| MET dose, /1000 patients | 16.1 | 14.6 | 18.0 | 16.4 | 14.8 | 16.9 | 17.8 | 16.1 | 18.3 | 15.5 | 0.603 |
| In-hospital mortality, /1000 patients | 9.20 | 10.10 | 9.64 | 8.13 | 8.14 | 8.37 | 8.10 | 8.26 | 7.97 | 7.23 | 0.004 |
| Admitted patients | 72,468 | 73,308 | 72,960 | 83,100 | 89,064 | 75,852 | 90,588 | 93,384 | 91,668 | 100,788 | <0.001 |

MET, medical emergency team.

Although there is no standard measure for evaluating the maturity of RRS from the existing literature on RRS maturity, several studies have revealed the correlation of the number of activation with the maturity of RRS [25–27]. In a long-term observational study, Herod R et al. found that progressively increased number of RRS activations concurred with lower hospital mortality [25]. Moriarty JP et al. also found that the reduction in rescue failure rates was associated with a substantial increase in the number of RRS activation [26]. In addition, timeless response to patient deterioration has been recommended as quality metrics of RRS process [9], since delayed activation of RRS is associated with higher in-hospital mortality [28,29]. In the present study, the increased number of the RRS activations and decreased the time from derangement to RRS activation concurred with lowered in-hospital mortality over 10 years, although no causality could be concluded. Therefore, it might be considered that our RRS has matured over the past decade.

Several previous studies have shown reduced incidence of IHCAs after the implementation and maturity of RRS [30,31]. However, these studies simply focused on reducing IHCAs regardless of the predictability of IHCA, although the overall rate of IHCAs might be limited for the evaluation of RRS [8]. Therefore, potentially preventable IHCAs, rather than total number of IHCAs, is recommended as a quality metric for the evaluation of RRS by the International Society of RRS [9]. In this study, the trend for the overall IHCA rate was stable with maturity of our RRS over 10 years. The lack of change in the overall rate of IHCAs might be associated with the number of I-IHCAs that were not suitable for resuscitation, which

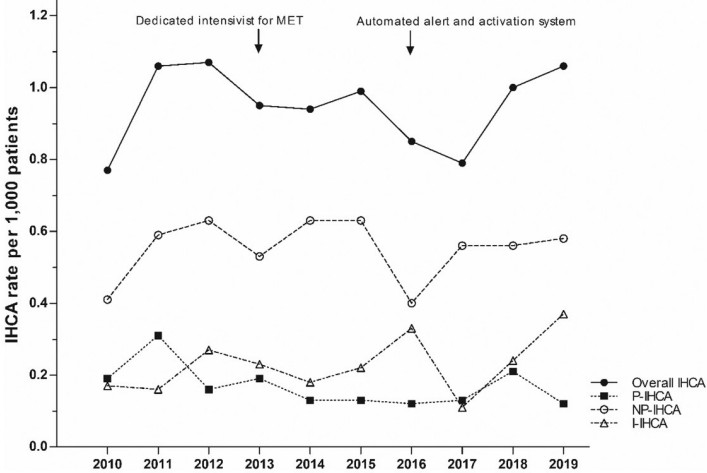

**Fig 2. Annual trend of in-hospital cardiac arrests (IHCAs) by classification.** P-IHCA, preventable in-hospital cardiac arrest; NP-IHCA, non-preventable in-hospital cardiac arrest; I-IHCA, inevitable in-hospital cardiac arrest.

contributed to the overall rate of IHCAs. Therefore, 23.2% of IHCAs might receive inappropriate CPR despite the futile situation reflecting both a low chance of survival and likely poorer quality of life afterward if spontaneous circulation is returned. This highlights a potential problem with using the overall rate of IHCAs as an outcome measure for RRS, which could explain the failure of previous studies to demonstrate consistently the efficacy of RRS in decreasing total hospital mortality [32].

Our results indicate that the most common circumstances of P-IHCAs were sudden critical illness in under-monitored patients and delays in initiating RRS response for monitored patients who met crisis criteria, which are consistent with previous reports [3,5–7]. Therefore, more IHCAs might be preventable by closer monitoring on floors and by preventing delays in addressing deterioration in patient condition. Effective risk management necessitates that preventable IHCA is minimized. Therefore, the effort for reducing preventable IHCA, rather than overall IHCA, could be a more appropriate quality metric to measure the clinical outcomes of RRS implementation and maturation; however, inconsistent definitions have limited its generalizability in a wide range of healthcare settings [9].

Although this study provides additional information on a more appropriate quality metric for RRS implementation and maturation using simple methods that are reproducible within the existing resources of most hospitals, there are several limitations that should be acknowledged. First, the study was limited by its inherent retrospective observational nature. However, data on treated IHCAs were prospectively collected from consecutive patients received CPRs. Therefore, our cohort is more likely to reflect the patients encountered in routine practice and thus can be readily applicable in similar settings. Second, the present study was conducted at a single institution with physician-based MET. Accordingly, our findings may have limited generalizability in other RRS. Finally, an automated alert and activation system was integrated into the original RRS activation process in August 2016. However, this change of activation process could be itself a sign of the maturation of our RRS (S1 Table).

## Conclusion

In conclusion, the incidence of P-IHCA could be a more appropriate quality metric to measure the clinical outcomes of RRS implementation and maturation than overall IHCA.

## Supporting information

**S1 Table. Comparison of MET activation, incidence of IHCAs, and in-hospital mortality before and after implementing the automated alert and activation system in August 2016.** (DOCX)

## Author Contributions

**Conceptualization:** Kyeongman Jeon.

**Data curation:** Hohyung Jung, Ryoung-Eun Ko, Myeong Gyun Ko.

**Formal analysis:** Hohyung Jung, Ryoung-Eun Ko, Kyeongman Jeon.

**Funding acquisition:** Kyeongman Jeon.

**Investigation:** Hohyung Jung, Ryoung-Eun Ko, Myeong Gyun Ko.

**Methodology:** Hohyung Jung, Ryoung-Eun Ko, Myeong Gyun Ko, Kyeongman Jeon.

**Supervision:** Kyeongman Jeon.

**Visualization:** Hohyung Jung, Ryoung-Eun Ko.

**Writing – original draft:** Hohyung Jung, Ryoung-Eun Ko, Kyeongman Jeon.

**Writing – review & editing:** Hohyung Jung, Ryoung-Eun Ko, Myeong Gyun Ko, Kyeongman Jeon.

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
