## [Decision Letter · Decision Letter 0]

27 Oct 2021

PONE-D-21-31238Trends of in-hospital cardiac arrests in a single tertiary hospital with a mature rapid response systemPLOS ONE

Dear Dr. Kyeongman Jeon,

Thank you for submitting your manuscript to PLOS ONE. After careful consideration, we feel that it has merit but does not fully meet PLOS ONE’s publication criteria as it currently stands. Therefore, we invite you to submit a revised version of the manuscript that addresses the points raised during the review process.

ACADEMIC EDITOR:

Thank you very much for having submitted this paper.  Although the paper could be of interest however there are several issues to be addressed. I hope the comments of the reviewers could serve you as a guide to improve the quality of the paper which in the present form is not suitable for publication.

We look forward to receiving your revised manuscript.

Kind regards,

Simone Savastano

Academic Editor

PLOS ONE

Journal Requirements:

Additional Editor Comments:

Thank you very much for having submitted this paper. Although the paper could be of interest however there are several issues to be addressed. I hope the comments of the reviewers could serve you as a guide to improve the quality of the paper which in the present form is not suitable for publication.

Reviewers' comments:

Reviewer's Responses to Questions

**Comments to the Author**

1. Is the manuscript technically sound, and do the data support the conclusions?

Reviewer #1: Partly

Reviewer #2: No

2. Has the statistical analysis been performed appropriately and rigorously? 

Reviewer #1: Yes

Reviewer #2: No

3. Have the authors made all data underlying the findings in their manuscript fully available?

Reviewer #1: Yes

Reviewer #2: Yes

4. Is the manuscript presented in an intelligible fashion and written in standard English?

Reviewer #1: Yes

Reviewer #2: Yes

5. Review Comments to the Author

Reviewer #1: Dear Editor

Dear Authors

This is a retrospective observational study focusing on the consequences of implementing a RRS on the incidence and distribution of IHCA in a major hospital. Of note, only patients who suffered of IHCA in the general wards were included, according to the quality metrics published by the third international consensus conference on RRS (2019), which is correctly cited in the manuscript (9). The authors analyzed the yearly distribution between I-IHCA, P-IHCA and NP-IHCA from the start (2010) to the end (2019) of the period in exam and found a statistically significant decrease in P-IHCA from 0.19/1000 patients in 2010 to 0.12/1000 patients in 2019, with a number of total IHCA stable through the years and consistent with the rest of the literature (around 1/1000 patients). Attributing this decrease to implementation and maturation of RRS, the authors conclude that the incidence of P-IHCA (as opposed to overall IHCA) may be a better indicator of the effects of the implementation and maturation of a RRS in a major hospital.

One of the strengths of the article is the number of the patients suffering from IHCA, unusually high for a study of this type, thanks to a decade-long thorough follow-up. Another point in favour of the research would be the prospectively collection of consecutive patients. It must also be noted that the article is well written, concise, with clear infographics and no major spelling errors (to my knowledge).

However, there are a few issues, stated below:

Major issues:

1) The definition of preventable IHCA used by the authors in the article is different from the one adopted in the 2019 consensus statement on RRS (9), first and foremost regarding the time window before the event of IHCA: “at least 30 min prior to and within 24” as stated by the consensus VS “from 8 hours to 30 minutes before arrest” as used by the authors in the article at line 96. Another (reported here for simplicity) minor issue on this definition is the diction of “preventable” IHCA used by the authors opposed to the one of “predictable” used in the consensus.

2) The results of the study presented in terms of reduction of P-IHCA between 2010 and 2019 as stated in line 160-161 “P-IHCA decreased from 0.19/1,000 patients in 2010 to 0.12/1,000 patients in 2019 (P = 0.006)” may be seen as misleading: examining FIGURE 2 one would note that the incidence of P-IHCA in 2018 was around 0.2/1000 patients (higher than 2010), hinting to the paradoxical conclusion that if the study would have stopped in 2018 it might have shown an increase in P-IHCA. It is also evident from the same table that in 2011 the incidence of P-IHCA was around 0.3/1000 patients (highest in the decade). Given the conclusion stated by the authors in line 208-209 “the incidence of P-IHCA could be a more appropriate quality metric to measure the clinical outcomes of RRS implementation and maturation than overall IHCA” (this is also the opinion of the reviewer on the matter), a single analysis of the incidence of P-IHCA in 2010 VS the incidence of P-IHCA in 2019 may not be seen by the reader as sufficient to back this conclusion. A suggestion to address this issue (and other minor issues stated below) would be to pool the data from 2010 to July 2016 vs August 2016 to 2019, the latter representing the period with a more experienced MET, with dedicated staff (since March 2013) and an automated activation system (since August 2016). If significant, this analysis would point toward the conclusion that the incidence of P-IHCA is reduced by the implementation and maturation of RRS and thus may be itself a more accurate quality metric than the incidence of overall IHCA.

Minor issues:

3) In line 45 “since March 2013 the MET was composed of dedicated intensivist physicians”: it is not stated what was the composition of the MET between 2010 and 2013.

4) A minor limitation of the study would be the switch in the activating process of the MET: the one stated in TABLE 1 until July 2016 vs the one based on MEWS since August 2016. One could argue that the adoption of the automated activation system based upon MEWS may be itself a sign of the “maturation” of the RRS: it could be wise in this case to present the results as before vs after the switch in the activation process (see more in comment 2).

5) It is not clear what was the survival at hospital discharge in the IHCA patients: was it 17.8% as stated in TABLE 2 or 31.5% as stated in line 149? In the same line the phrase “Among 252 patients (31.5%) survived at discharge and only 111 (13.9%) patients…” probably “and” is a typo

Great work, best regards

Reviewer #2: This paper may be of interest as it highlights the importance of MET in the treatment oif cardiac arrest in a big hospital. However, it is a description of MET activity over a time period and it fails to reach conclusions regarding som of the potentially more interesting aspects including the DNACPR Decisions.“

Additionally some data are reported without supporting the causality of those with the main topic of the paper. I.g.

Authors report the in-hospital mortality rate was significantly decreased (from 9.20/1,000 patients in 2010 to 7.23/1,000 patients in 2019). What this due to MEt "maturity"? If this is the underling message, there are not data supporting it.

The definition of NP IHCA is quite wide additionally it is unclear whether the occurrence within 30 minutes form MET activation was also related to a delay in MET arrival at bedside considering that the goal is the get the MET within 10 minutes.

Which is the clinical message of the following sentence: Among 252 patients (31.5%) survived at discharge and only 111 (13.9%) patients were 150 managed by our MET within 24 hours before the event

Discussion section should be widen

6. PLOS authors have the option to publish the peer review history of their article (what does this mean?). If published, this will include your full peer review and any attached files.

Reviewer #1: **Yes: **Alessandro Fasolino

Reviewer #2: No

---

## [Author Response · Author response to Decision Letter 0]

24 Nov 2021

Response to Reviewers’ Comments

Reviewer#1:

This is a retrospective observational study focusing on the consequences of implementing a RRS on the incidence and distribution of IHCA in a major hospital. Of note, only patients who suffered of IHCA in the general wards were included, according to the quality metrics published by the third international consensus conference on RRS (2019), which is correctly cited in the manuscript (9). The authors analyzed the yearly distribution between I-IHCA, P-IHCA and NP-IHCA from the start (2010) to the end (2019) of the period in exam and found a statistically significant decrease in P-IHCA from 0.19/1000 patients in 2010 to 0.12/1000 patients in 2019, with a number of total IHCA stable through the years and consistent with the rest of the literature (around 1/1000 patients). Attributing this decrease to implementation and maturation of RRS, the authors conclude that the incidence of P-IHCA (as opposed to overall IHCA) may be a better indicator of the effects of the implementation and maturation of a RRS in a major hospital. One of the strengths of the article is the number of the patients suffering from IHCA, unusually high for a study of this type, thanks to a decade-long thorough follow-up. Another point in favour of the research would be the prospectively collection of consecutive patients. It must also be noted that the article is well written, concise, with clear infographics and no major spelling errors (to my knowledge).

: The authors really appreciate the reviewer’s encouraging comments. Regarding to your comments, we revised our manuscript based on your comments and suggestions.

Comments 1 (C1). The definition of preventable IHCA used by the authors in the article is different from the one adopted in the 2019 consensus statement on RRS (9), first and foremost regarding the time window before the event of IHCA: “at least 30 min prior to and within 24” as stated by the consensus VS “from 8 hours to 30 minutes before arrest” as used by the authors in the article at line 96. Another (reported here for simplicity) minor issue on this definition is the diction of “preventable” IHCA used by the authors opposed to the one of “predictable” used in the consensus.

Response 1 (R1). We thank the reviewer for valuable comments. Although, this study was conducted to identify the rates and characteristics of IHCAs according to the quality metrics published by the third international consensus conference on RRS, but, we used the terms of ‘predictability’ which used in the study for mature rapid response system conducted by MERIT committee (1). In this study, predictability was defined as: The events were termed “predictable” if the patient chart indicated objective or clear evidence of patient deterioration in the 6‐h period before the arrest event. In our hospital setting, patients admitted to the general ward checked vital signs at least every 8 hours. Therefore, preventable IHCA was defined as a cardiac arrest with preexisting signs of acute physiologic disturbance that fulfilled the MET activation criteria from 8 hours to 30 minutes before arrest. After reviewing several terms for the definition including avoidable, preventable, and predictable, ‘preventable IHCA’ was selected after discussing with the team members of our MET.

Reference:

1. Galhotra S, DeVita MA, Simmons RL, Dew MA; Members of the Medical Emergency Response Improvement Team (MERIT) Committee. Mature rapid response system and potentially avoidable cardiopulmonary arrests in hospital. Qual Saf Health Care. 2007 Aug;16(4):260-5.

C2. The results of the study presented in terms of reduction of P-IHCA between 2010 and 2019 as stated in line 160-161 “P-IHCA decreased from 0.19/1,000 patients in 2010 to 0.12/1,000 patients in 2019 (P = 0.006)” may be seen as misleading: examining FIGURE 2 one would note that the incidence of P-IHCA in 2018 was around 0.2/1000 patients (higher than 2010), hinting to the paradoxical conclusion that if the study would have stopped in 2018 it might have shown an increase in P-IHCA. It is also evident from the same table that in 2011 the incidence of P-IHCA was around 0.3/1000 patients (highest in the decade). Given the conclusion stated by the authors in line 208-209 “the incidence of P-IHCA could be a more appropriate quality metric to measure the clinical outcomes of RRS implementation and maturation than overall IHCA” (this is also the opinion of the reviewer on the matter), a single analysis of the incidence of P-IHCA in 2010 VS the incidence of P-IHCA in 2019 may not be seen by the reader as sufficient to back this conclusion. A suggestion to address this issue (and other minor issues stated below) would be to pool the data from 2010 to July 2016 vs August 2016 to 2019, the latter representing the period with a more experienced MET, with dedicated staff (since March 2013) and an automated activation system (since August 2016). If significant, this analysis would point toward the conclusion that the incidence of P-IHCA is reduced by the implementation and maturation of RRS and thus may be itself a more accurate quality metric than the incidence of overall IHCA.

R2. We thank the reviewer for valuable comments. First of all, the figure 2 may not clearly show the change over time of each group, which may cause confusion for readers. Therefore, line chart in the same manner as figure 1 would be more appropriate. Second, the annual changes of IHCA rates over the study period were analyzed with tests for trend, not just acomparison of arithmetic values between 2010 and 2019, as decribed in the Method section. However, we really appreciate your valuable suggestion for additional comparison of the pool data before and after the switch in the activating process of our MET. As expected, the rate of P-IHCA was reduced after implementing the aumated alert and activation system from 0.192/1000 patients to 0.124/1000 patietns, while the number of MET activation increased and in-hospital mortality decreased. We added this information in the Discussion section and provided as a supplemental table (S1 Table in the supporting information file). 

C3. In line 45 “since March 2013 the MET was composed of dedicated intensivist physicians”: it is not stated what was the composition of the MET between 2010 and 2013.

R3. We thank the reviewer for valuable comments. At the beginning, the team was initially consisted of either fellows that were training in critical care or senior residents in internal medicine. We added this information in the Method section of the revised manuscript.

C4. A minor limitation of the study would be the switch in the activating process of the MET: the one stated in TABLE 1 until July 2016 vs the one based on MEWS since August 2016. One could argue that the adoption of the automated activation system based upon MEWS may be itself a sign of the “maturation” of the RRS: it could be wise in this case to present the results as before vs after the switch in the activation process (see more in comment 2).

R4. This is an excellent point of view. We totally agree that the adoption of the automated alert and activation system could be itself a sign of the “maturation” of the RRS. We added this information in the Discussion section and the results from additional comparison of the pool data before and after the system would be provided as a supplemental table (S1 Table in the supporting information file). 

C5. It is not clear what was the survival at hospital discharge in the IHCA patients: was it 17.8% as stated in TABLE 2 or 31.5% as stated in line 149? In the same line the phrase “Among 252 patients (31.5%) survived at discharge and only 111 (13.9%) patients…” probably “and” is a typo.

R5. We apologize for our carelessness. We fixed the error in the sentence and modified to clarify the meaning of the sentence.

Reviewer#2:

This paper may be of interest as it highlights the importance of MET in the treatment of cardiac arrest in a big hospital. However, it is a description of MET activity over a time period and it fails to reach conclusions regarding som of the potentially more interesting aspects including the DNACPR Decisions.“

: We understand the reviewer’s concern that this study focused only on the consequences of implementing a RRS on the incidence and distribution of IHCA. However, we strive to analyze the number of patients suffering from IHCA over a 10-year period, and aimed to identify the rates and characteristics of IHCAs including predictability of the IHCAs as our RRS matures. Regarding to your comments, we revised our manuscript based on your comments and suggestions.

Comments 1 (C1). Additionally some data are reported without supporting the causality of those with the main topic of the paper. I.g. Authors report the in-hospital mortality rate was significantly decreased (from 9.20/1,000 patients in 2010 to 7.23/1,000 patients in 2019). What this due to MET "maturity"? If this is the underling message, there are not data supporting it.

Response 1 (R1). We thank the reviewer for valuable comments. Previous studies have attempted to assess the effect of the RRS with a change of hospital mortality, and meta-analyses have shown that RRS might be associated with reduced in-hospital mortality and IHCA. Therefore, the change of hospital mortality in our hospital could be considered to be related with the implantation and maturity of MET over ten years. However, unfortunately, there is no standard measure for evaluating the maturity of RRS. Several studies have revealed the correlation of the number of activation with the maturity of RRS. In addition, delayed activation is associatd with higher in-hospital mortality so that timeless response to patient deterioration has been recommended as quality metrics of RRS process. In this study, the increased number of the MET activations and decreased the time from derangement to MET activation concurred with lowered in-hospital mortality over 10 years, although no causality could be concluded. Therefore, it might be considered that our MET has matured over the past decade. We added this point in the Discussion section.

C2. The definition of NP IHCA is quite wide additionally it is unclear whether the occurrence within 30 minutes form MET activation was also related to a delay in MET arrival at bedside considering that the goal is the get the MET within 10 minutes.

R2. We thank the reviewer for valuable comments. Previous studies showed that activation of the RRS in the presence of objective escalation criteria in a period of more than 30 minutes prior to an IHCA may allow the RRS to prevent the event from occurring. Periods of less than 30 minutes may not be sufficient to allow the RRS to effectively intervene. Therefore, we defined NP IHCA as a cardiac arrest that occurred within 8 hours after admission, or without any record of vital signs within 8 hours before arrest, or within 30 minutes after drug administration or procedures, or from unexpected lethal arrhythmia; this includes cardiac arrest that occurred within 30 minutes after MET activation.

Reference:

1. Downey AW, Quach JL, Haase M, Haase-Fielitz A, Jones D, Bellomo R. Characteristics and outcomes of patients receiving a medical emergency team review for acute change in conscious state or arrhythmias. Crit Care Med. 2008 Feb;36(2):477-81.

2. Quach JL, Downey AW, Haase M, Haase-Fielitz A, Jones D, Bellomo R. Characteristics and outcomes of patients receiving a medical emergency team review for respiratory distress or hypotension. J Crit Care. 2008 Sep;23(3):325-31.

C3. Which is the clinical message of the following sentence: Among 252 patients (31.5%) survived at discharge and only 111 (13.9%) patients were managed by our MET within 24 hours before the event.

R3. We apologize for lack of clarity. We intended to show that the majority of patients suffered form IHCA was not managed by MET before the IHCA event, even in patient survived to discharge from the hospital. But, there was a typo in the sentence (there should have been no 'and’), which made it fail to convey meaning. We corrected the error in the revised manuscript.

C4. Discussion section should be widen.

C4. We added the discussion of the MET maturity in the Discussion section of the revised manuscript.

---

## [Decision Letter · Decision Letter 1]

29 Dec 2021

Trends of in-hospital cardiac arrests in a single tertiary hospital with a mature rapid response system

PONE-D-21-31238R1

Dear Dr. Jeon,

We’re pleased to inform you that your manuscript has been judged scientifically suitable for publication and will be formally accepted for publication once it meets all outstanding technical requirements.

Kind regards,

Simone Savastano

Academic Editor

PLOS ONE

Additional Editor Comments (optional):

Thank you very much for having addressed properly the comments of the reviewers. Now the paper has gain in quality and clarity so it could be suitable for publication.

Reviewers' comments:

Reviewer's Responses to Questions

**Comments to the Author**

1. If the authors have adequately addressed your comments raised in a previous round of review and you feel that this manuscript is now acceptable for publication, you may indicate that here to bypass the “Comments to the Author” section, enter your conflict of interest statement in the “Confidential to Editor” section, and submit your "Accept" recommendation.

Reviewer #1: All comments have been addressed

Reviewer #2: All comments have been addressed

2. Is the manuscript technically sound, and do the data support the conclusions?

Reviewer #1: Yes

Reviewer #2: Yes

3. Has the statistical analysis been performed appropriately and rigorously? 

Reviewer #1: Yes

Reviewer #2: Yes

4. Have the authors made all data underlying the findings in their manuscript fully available?

Reviewer #1: Yes

Reviewer #2: Yes

5. Is the manuscript presented in an intelligible fashion and written in standard English?

Reviewer #1: Yes

Reviewer #2: Yes

6. Review Comments to the Author

Reviewer #1: Dear Editor, dear Authors

my suggestions in the last review were sufficiently addressed by the authors.

One last minor issue lies in the comment. The authors replied in R1 "In our hospital setting, patients admitted to the general ward checked vital signs at least every 8 hours", but in the Method section - operation of RRS it was stated "The frequency of measuring vital signs was made according to the order of the attending physician, but vital signs were usually measured at least four times a day and more often when the patient’s clinical condition changed". Please be more clear on how frequently the minimum of vital signs are collected.

Reviewer #2: The author have imporved the information of the paper. I have no further comments to be addressed.

7. PLOS authors have the option to publish the peer review history of their article (what does this mean?). If published, this will include your full peer review and any attached files.

Reviewer #1: **Yes: **Alessandro Fasolino

Reviewer #2: No

---

## [Editor Report · Acceptance letter]

6 Jan 2022

PONE-D-21-31238R1 

Trends of in-hospital cardiac arrests in a single tertiary hospital with a mature rapid response system 

Dear Dr. Jeon:

I'm pleased to inform you that your manuscript has been deemed suitable for publication in PLOS ONE. Congratulations! Your manuscript is now with our production department. 

Kind regards, 

on behalf of

Dr. Simone Savastano 

Academic Editor

PLOS ONE